# Observation of nuclear-spin Seebeck effect

T. Kikkawa [1,2,3 ✉], D. Reitz[4], H. Ito[1], T. Makiuchi [1], T. Sugimoto[1], K. Tsunekawa[1], S. Daimon [1], K. Oyanagi [3,5], R. Ramos [2,9], S. Takahashi[2], Y. Shiomi[6], Y. Tserkovnyak [4] & E. Saitoh[1,2,3,7,8]

Thermoelectric effects have been applied to power generators and temperature sensors that convert waste heat into electricity. The effects, however, have been limited to electrons to occur, and inevitably disappear at low temperatures due to electronic entropy quenching. Here, we report thermoelectric generation caused by nuclear spins in a solid: nuclear-spin Seebeck effect. The sample is a magnetically ordered material $MnCO_3$ having a large nuclear spin ($I = 5/2$) of $^{55}Mn$ nuclei and strong hyperfine coupling, with a Pt contact. In the system, we observe low-temperature thermoelectric signals down to 100 mK due to nuclear-spin excitation. Our theoretical calculation in which interfacial Korringa process is taken into consideration quantitatively reproduces the results. The nuclear thermoelectric effect demonstrated here offers a way for exploring thermoelectric science and technologies at ultralow temperatures.

[1] Department of Applied Physics, The University of Tokyo, Tokyo, Japan. [2] WPI Advanced Institute for Materials Research, Tohoku University, Sendai, Japan. [3] Institute for Materials Research, Tohoku University, Sendai, Japan. [4] Department of Physics and Astronomy, University of California, Los Angeles, CA, USA. [5] Faculty of Science and Engineering, Iwate University, Morioka, Japan. [6] Department of Basic Science, The University of Tokyo, Tokyo, Japan. [7] Advanced Science Research Center, Japan Atomic Energy Agency, Tokai, Japan. [8] Institute for AI and Beyond, The University of Tokyo, Tokyo, Japan. [9] Present address: Centro de Investigación en Química Biolóxica e Materiais Moleculares (CIQUS), Departamento de Química-Física, Universidade de Santiago de Compostela, Santiago de Compostela, Spain. ✉email: t.kikkawa@ap.t.u-tokyo.ac.jp

Thermoelectric effects enable the direct conversion of thermal energy into electric energy, promising for power generation and waste heat recovery. Most of the prevalent thermoelectric generators have relied on the Seebeck effect, which is the generation of an electric voltage by placing a conductor junction in a temperature gradient[1–3]. Recently, in the study of spintronics, a spin analog of the Seebeck effect—the spin Seebeck effect (SSE)[4–20]—was discovered. The SSE is the generation of a spin current, a flow of spin angular momentum, as a result of a temperature gradient applied across a junction consisting of a magnet and a metal[17]. In electronic SSE, a thermally generated magnon flow in a magnet injects a conduction-electron spin current into the adjacent metal via the interfacial electronic spin exchange[8,9,13,16]. The spin current injected into a metal can be converted into a voltage by the inverse spin Hall effect (ISHE)[21–24], enabling unexplored approaches toward thermoelectric conversion and energy-harvesting technologies[10,17,18].

Up to now, all the thermoelectric effects have been an exclusive feature of electrons[1–20]. At low temperatures, however, their efficiency is dramatically suppressed, as the thermodynamic entropy of electrons steeply reduces to zero when approaching absolute zero temperature, according to the third law of thermodynamics. In the case of Seebeck effects in semiconductors, the entropy reduction is related to the exponential suppression of the thermally excited charge carriers[2], whereas, in SSEs, it is related to the freezing out of spin fluctuations (magnons)[15–17]. Seebeck effects in metals are also suppressed at low temperatures, as the efficiency is governed by $k_B T / \epsilon_F$[1], where $k_B$ is the Boltzmann constant, $T$ the environmental temperature, and $\epsilon_F$ the Fermi energy. Therefore, so far, the thermoelectric applications have been limited to higher temperatures, as no mechanism in the ultralow temperature regime (~mK range) has been found.

In solids, there is a hitherto unexplored entropy carrier that is well activated even at ultralow temperatures: a nuclear spin. Because of its tiny gyromagnetic ratio $\gamma_n$ (~$10^3$ times less than that of electrons[25] $\gamma_e$), a nuclear spin exhibits much lower excitation energy than that of electron spins in ambient fields, allowing its thermal agitation. Here, a question arises: can nuclear spins generate thermoelectric effects? If spin angular momentum can be extracted from nuclei in the form of an electron spin current under a temperature bias, it should generate a thermoelectric voltage via the ISHE in an attached metal, realizing all-solid-state thermoelectricity based on atomic nuclei.

Here we report an observation of the nuclear SSE (Fig. 1a) in a heterostructure composed of a Pt film and a crystal of easy-plane canted antiferromagnetic MnCO$_3$[26–28] (Fig. 1d). In MnCO$_3$, $^{55}$Mn nuclei, a 100% natural-abundance isotope, carry a large spin $I$ of 5/2 and exhibit strong hyperfine coupling with electrons, which allows spin transfer between nuclei and electrons as recently found in the spin pumping measurements under nuclear magnetic resonance[28]. In MnCO$_3$ single crystals covered with Pt films, we found a strong thermoelectric signal enhancement down to 100 mK (Fig. 1e), as shown below, which demonstrates thermoelectric generation at ultralow temperatures. The experimental results are quantitatively reproduced by a theory for nuclear SSE in which the Korringa process[29] due to the hyperfine coupling between nuclear spins in the MnCO$_3$ and conduction-electron spins in the attached Pt is taken into consideration (Fig. 1a).

## Results

### Sample and measurement setup

We have used the ISHE[21–24] in the Pt film to detect a spin current injected into the film. The ISHE converts a spin current, $\mathbf{J}_s$, into an electric field, $\mathbf{E}_{ISHE}$, through the spin–orbit interaction of conduction electrons, which can be strong in heavy metals such as Pt[10,17]. When a spin

current induced by a nuclear SSE carries spin polarization $\hat{\mathbf{s}}$ parallel to the net nuclear-spin polarization $\mathbf{I}$ along the spatial direction $\mathbf{J}_s$, $\mathbf{E}_{ISHE}$ is given by (Fig. 1b)

$$\mathbf{E}_{ISHE} = \frac{2e}{\hbar}\rho\theta_{SHE}\mathbf{J}_s \times \hat{\mathbf{s}} \qquad (1)$$

where $\rho$ and $\theta_{SHE}$ are the resistivity and the spin Hall angle of the Pt layer, respectively. By measuring $\mathbf{E}_{ISHE}$, nuclear SSEs can be detected electrically. We note that, as the spin current $\mathbf{J}_s$ flows normal to the Pt/MnCO$_3$ interface ($\mathbf{J}_s \parallel \mathbf{x}$), the resultant voltage signal $V$ is maximal for $\hat{\mathbf{s}}(\parallel\mathbf{I}) \parallel \mathbf{z}$, when $\mathbf{E}_{ISHE}$ is measured along the $\mathbf{y}$ direction shown in Fig. 1b. However, because of the tiny Zeeman coupling of nuclear spins, it is challenging to control nuclear-spin polarization by using $\mathbf{B}$, unlike electronic magnetization in conventional magnets. Nevertheless, we can overcome the difficulty by using a magnetic ordered material carrying a large nuclear spin and strong hyperfine coupling. We have noticed that an antiferromagnet MnCO$_3$ ($I = 5/2$)[26–28] satisfies all such conditions. Below the Néel temperature ($T_N = 35$ K) of MnCO$_3$, the Mn$^{2+}$ sublattice magnetizations $\mathbf{M}_1$ and $\mathbf{M}_2$ are aligned in the (111) plane and canted slightly from the collinear antiferromagnetic configuration due to the bulk Dzyaloshinskii–Moriya interaction[26] (see Fig. 1d and Supplementary Note 1). The hyperfine (Overhauser) fields $\mathbf{B}_{hf}$ acting on the $^{55}$Mn sublattice nuclear spins $\mathbf{I}_1$ and $\mathbf{I}_2$ due to $\mathbf{M}_1$ and $\mathbf{M}_2$ reach as large as 57 T[28], which induce nuclear-spin polarization (~40% at 100 mK) and orient $\mathbf{I}_1$ and $\mathbf{I}_2$ along the $\mathbf{M}_1$ and $\mathbf{M}_2$ directions, respectively[30], as shown in Fig. 1d. Moreover, the net nuclear-spin polarization ($\mathbf{I}_1$ and $\mathbf{I}_2$) direction can be controlled by applying $\mathbf{B}$, as the canting angle $\theta$ of $\mathbf{M}_1$ and $\mathbf{M}_2$ changes with $\mathbf{B}$, owing to the very weak magnetocrystalline anisotropy (~0.1 mT within the easy plane[26], see Fig. 1d). The advantage enables us to prepare a controllable nuclear-spin polarization in MnCO$_3$, making nuclear SSE experiments feasible.

The SSE devices used in the present study consist of a 10 nm-thick Pt strip [200 μm long ($l$) and 100 nm wide ($w$)] deposited on the top of an insulating MnCO$_3$ (111) ($3 \times 3 \times 0.5$ mm$^3$) crystal (see "Methods" and Supplementary Note 2). The Pt strip acts as a heater as well as a spin-voltage converter based on the ISHE for measuring nuclear SSEs: by applying an a.c. current $I_c$ ($= \sqrt{2}I_{rms} \sin \omega t$) to the Pt strip to generate heat and measuring the second harmonic voltage $V$ generated in the Pt by a lock-in technique[11,14], we can selectively detect the ISHE voltage arising from the temperature drop across the Pt/MnCO$_3$ interface induced by the Joule heating $\propto I_{rms}^2$ of the applied current. The SSE experiments were conducted with a $^4$He cryostat down to 1.82 K using a Pt/MnCO$_3$ device named Device 1 and with a $^3$He–$^4$He dilution refrigerator down to 100 mK using a similar device named Device 2. The Pt/MnCO$_3$ devices were mounted in the cryostats and the magnetic field $\mathbf{B}$ was applied along the $\mathbf{z}$ direction as shown in Fig. 1b. Further details are described in "Methods."

### Observation of nuclear SSE

In Fig. 2a, we show the voltage $V$ data measured at $T = 20$ K and 1.82 K for the Pt/MnCO$_3$ Device 1. At 20 K, no voltage signal appears with the application of $B$. On the other hand, at a lower temperature $T = 1.82$ K, an unconventional voltage signal shows up. The sign of $V/I_{rms}^2$ reverses by reversing the $B$ direction. The signal intensity increases monotonically with increasing $B$ from zero and it takes a broad peak at around 4 T. For further high $B$, $V/I_{rms}^2$ starts to decrease. We confirmed that the observed signal shares the characteristic feature of ISHE induced by SSE[10,17]; $V$ appears only when a heat current is applied and the $V$ intensity scales linearly with the heat power $\propto I_{rms}^2$. The signal intensity is maximal when $\mathbf{B} \parallel \mathbf{z}$ but

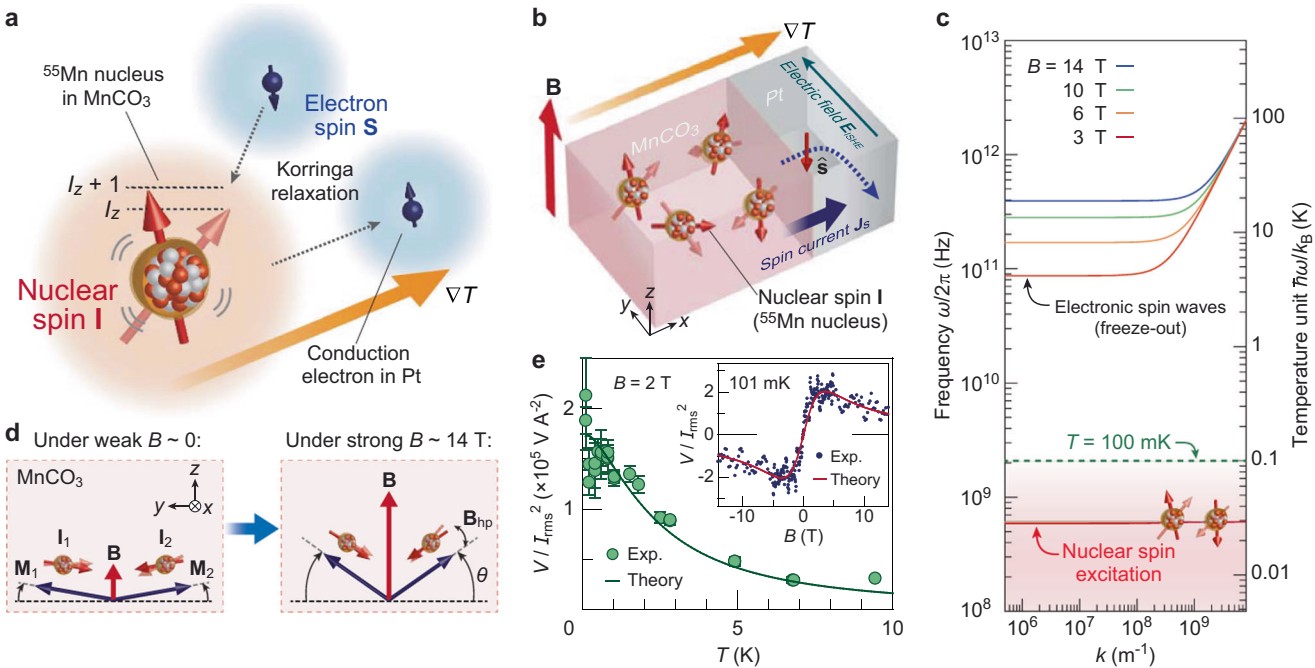

**Fig. 1 Concept of nuclear-spin Seebeck effect in Pt/MnCO₃. a** Schematic illustration of the nuclear SSE induced by the Korringa relaxation process[29], the spin-conserving flip-flop scattering between a nuclear spin, **I**, of $^{55}$Mn in MnCO₃ and an electron spin, **S**, in Pt via the interfacial hyperfine interaction. $I_z$ represents the $z$ component of the nuclear spin **I**. **b** Schematic illustration of the nuclear SSE and the ISHE in a Pt/MnCO₃ hybrid structure, where MnCO₃ contains nuclear spin $I = 5/2$ on $^{55}$Mn (100% natural abundance). When a temperature gradient ($\nabla T$) is applied across the Pt/MnCO₃ interface, a spin current ($\mathbf{J_s}$) carrying a spin polarization vector $\hat{s}$ is induced in the Pt layer by the nuclear SSE, which is subsequently converted into an electric field ($\mathbf{E_{ISHE}}$) via the ISHE in the direction of $\mathbf{J_s} \times \hat{s}$[22]. Here, $\hat{s}$ is along the external magnetic field **B**. **c** A calculated dispersion relation of the electronic spin wave and energy of the nuclear-spin excitation in MnCO₃ at a temperature $T$ of 100 mK for several magnetic fields[26–28]. The energy level of 100 mK is plotted with a green dashed line. At $T = 100$ mK, the electronic spin waves are frozen out, whereas nuclear spins remain thermally active. **d** Schematic illustration of the orientation of the Mn$^{2+}$ sublattice electronic magnetization $\mathbf{M_1}$ and $\mathbf{M_2}$, and the $^{55}$Mn nuclear spins $\mathbf{I_1}$ and $\mathbf{I_2}$ in MnCO₃ in the (111) plane when the external field **B** is applied in the plane (**B** ∥ **z**). Below the antiferromagnetic ordering temperature $T_N = 35$ K of MnCO₃, $\mathbf{M_1}$ and $\mathbf{M_2}$ are aligned in the (111) plane and canted slightly from the pure antiferromagnetic ordering direction due to the bulk Dzyaloshinskii–Moriya interaction[26] (Supplementary Fig. 1). The canting angle $\theta$ increases with the external field. $\theta = 0.26°$ at zero field, whereas $\theta = 12°$ at $B = 14$ T. Due to the strong hyperfine (Overhauser) field of $B_{hf} \sim 57$ T, the sublattice nuclear spins $\mathbf{I_1}$ and $\mathbf{I_2}$ orient antiparallel to the electronic magnetization $\mathbf{M_1}$ and $\mathbf{M_2}$ directions, respectively. Here, the antiparallel orientation originates from the nature of the contact hyperfine interaction and the relative sign of the nuclear and electronic gyromagnetic ratios $\gamma_n$ and $\gamma_e$[30]. **e** Experimental demonstration of the nuclear SSE in Pt/MnCO₃. Temperature ($T$) dependence of the thermoelectric voltage $V$ (normalized by the applied heat power $\propto I^2_{rms}$) at $B = 2$ T. The error bar represents the SD. The inset shows the $B$ dependence of $V/I^2_{rms}$ at $T = 101$ mK. Theoretical results for the nuclear SSE are also plotted with solid curves.

vanishes when **B** ⊥ **z**, consistent with the prediction of Eq. (1). The sign of $V$ reverses when the Pt strip ($\theta_{SHE} > 0$) is replaced with tungsten exhibiting a negative[10] $\theta_{SHE}$. The results confirm that the voltage signal is induced by thermally driven spin currents and ISHE (see Supplementary Notes 3–5 for details).

Surprisingly, the signal intensity persists down to the ultralow temperature regime. Figure 2c, d show the $B$ dependence of $V/I^2_{rms}$ at 1.8 K < $T$ < 50 K for Device 1 and at 100 mK < $T$ < 1.6 K for Device 2, respectively. With decreasing temperature $T$ starting from 50 K, the SSE signal appears below ~10 K and its intensity dramatically increases by further decreasing $T$ (see Fig. 2c and b, in which the $T$ dependence of the maximum $V/I^2_{rms}$ is plotted). Importantly, the signal intensity continues to increase down to ultralow temperatures on the order of ~100 mK (see Fig. 2d and the inset to Fig. 2b). Moreover, the signal persists in the higher field range up to 14 T even at such ultralow temperatures, which is totally distinct from the conventional SSE driven by electronic magnetization dynamics. For instance, in ferrimagnetic Y₃Fe₅O₁₂, the SSE intensity decreases monotonically with decreasing temperature below 20 K and completely disappears below 5 K at 14 T due to the freezing out of magnons[15–17]. The maximum output of Device 2 normalized by its electrical resistance $R_{Pt}$, heating power $R_{Pt}I^2_{rms}$, and geometric factor $l^{-1}$ is as large as

$V_{max}l/(R^2_{Pt}I^2_{rms}) \sim 58$ nA mW$^{-1}$ at 101 mK, which is nearly two orders of magnitude higher than that of a prototypical room-temperature SSE device made of Pt/Y₃Fe₅O₁₂ (~1 nA mW$^{-1}$) having the same electrode and heater dimensions [see Supplementary Note 7 and Eq. (3) in Supplementary Note 9 for details].

**Nuclear- and electron-spin excitation spectra in MnCO₃.** We now discuss the results in terms of the nuclear- and electron-spin excitation features in MnCO₃. In Fig. 1c, we show the electronic and nuclear-spin excitation spectra in MnCO₃[26–28] for several fields at $T = 100$ mK, whose thermal energy $k_B T$ is depicted as the green dashed line. Above $k_B T$, thermal excitation is exponentially suppressed. The lower branch at around 600 MHz, corresponding to 30 mK, originates from the nuclear-spin excitation $\omega_n$, whose excitation gap is dominated by the strong hyperfine internal field $B_{hf} = \omega_n/\gamma_n \sim 57$ T. The upper branches represent the electronic spin-wave modes $\omega_{mk}$, which shift toward higher frequencies with increasing $B$ due to the strong Zeeman effect. At $B = 14$ T, the electronic spin excitation gap $\omega_{m0}(\approx \gamma_e B)$ is ~19 K, two orders of magnitude greater than the thermal energy = 100 mK, resulting in a negligibly small value of the Boltzmann factor $\exp(-\hbar\omega_{m0}/k_B T) \sim 10^{-82} \ll 1$. If the SSE we measured were driven by the electronic spin-wave modes, the SSE signal would

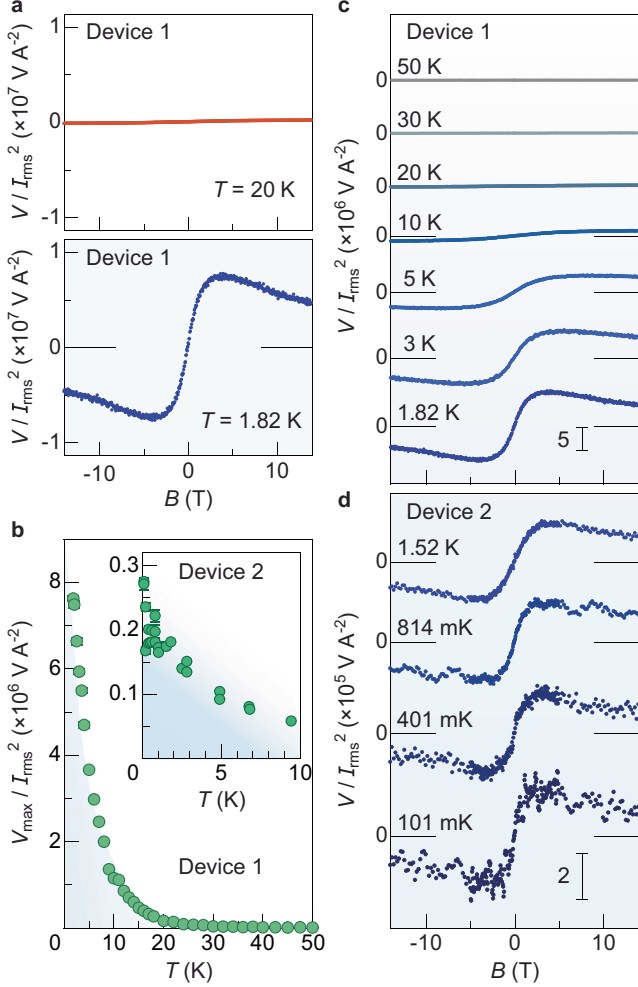

**Fig. 2 Observation of nuclear-spin Seebeck effect in Pt/MnCO₃. a** $B$ dependence of $V/I^2_{rms}$ (voltage $V$ normalized by the square of the applied charge current $I_{rms}$) for the Pt/MnCO₃ Device 1 at $T = 20$ K (red) and 1.82 K (blue). **b** $T$ dependence of the maximum $V/I^2_{rms}$ (defined as $V_{max}/I^2_{rms}$) for the Pt/MnCO₃ Device 1 at $1.8$ K $< T < 50$ K. The inset shows the $T$ dependence of $V_{max}/I^2_{rms}$ for the Pt/MnCO₃ Device 2 at 100 mK $< T < 10$ K measured with a dilution refrigerator. The error bar represents the SD. **c, d** $B$ dependence of $V/I^2_{rms}$ for the Pt/MnCO₃ Devices 1 (**c**) and 2 (**d**) at selected temperatures. The Pt/MnCO₃ Device 1 exhibits electrical resistance one order of magnitude higher than that for Device 2, resulting in an overall higher intensity of $V/I^2_{rms}$ in the Pt/MnCO₃ Device 1 (see Supplementary Note 6 for details).

be completely suppressed by applying a strong field of 14 T, as with the conventional SSE of Y₃Fe₅O₁₂[15]. This clearly shows the irrelevance of the electronic SSE to the observed signal at low temperatures. On the other hand, the nuclear-spin mode can be greatly excited even by such a small thermal energy of ~100 mK and it remains almost unaffected by the applied $B$ due to the tiny Zeeman effects, much weaker than the hyperfine internal field ~ 57 T (Fig. 1c); the nuclear spins can contribute to SSEs even in such a low-$T$ and high-$B$ environment. The results also suggest that direct coupling between nuclear spins in the MnCO₃ and electrons in the Pt at the interface should be responsible for the SSE, rather than the interfacial electronic exchange mediated by the gapped magnons under strong magnetic fields.

**Theoretical model for nuclear SSE.** We theoretically model the nuclear SSE in which direct nuclear-electron coupling due to the

Korringa process[29] is taken into consideration. In the model, the spin current $J_{ne}$ is generated by the interfacial hyperfine interaction between nuclear spins in the MnCO₃ and conduction-electron spins in the Pt under the temperature bias $T_e - T_p$ (see Fig. 3a and Supplementary Note 9 for details). Here, $T_e$ and $T_p$ represent effective temperatures for electrons in the Pt and phonons in the MnCO₃ near the interface, respectively. The nuclear-spin current $J_{ne}$ arises in proportion to the effective temperature difference between the electrons in the Pt ($T_e$) and nuclei in the MnCO₃ ($T_n$): $J_{ne} = \Gamma_{ne}k_B(T_e - T_n)$. Here $T_n$ may deviate from the electron $T_e$ due to the nuclear-phonon thermalization in MnCO₃ given by $J_{np} = \Gamma_{np}k_B(T_n - T_p)$, resulting in the finite spin current $J_{ne}$. The expression for the nuclear SSE coefficient reads

$$\mathcal{S}_n = \frac{g_n^{\uparrow\downarrow}}{4\pi I}\pi\chi b \frac{\hbar\omega_n}{k_B T}\left[\frac{T_e - T_n}{T_e - T_p}\right] \qquad (2)$$

where $g_n^{\uparrow\downarrow}$ is the nuclear spin-mixing conductance per unit area, $\chi$ the normalized antiferromagnetic transverse susceptibility such that $\theta = \chi b$ is the canting angle, $b \equiv \hbar\gamma_e s_e B$ the normalized magnetic field with saturated spin density $s_e$ ($s_e \equiv S/V$, for volume per site $V$), and $T$ the average temperature. The bracketed expression in Eq. (2) is evaluated as $(T_e - T_n)/(T_e - T_p) = (1 + \Gamma_{ne}/\Gamma_{np})^{-1}$ from the steady-state condition $J_{ne} = J_{np}$[31]. Here, $\Gamma_{ne} \propto 1/T$ [Eq. (1) shown in Supplementary Note 9] and $\Gamma_{np} \propto 1/T\omega^2_{m0}$ is derived by Fermi's Golden rule for the nuclear-phonon thermalization rate mediated by virtual magnons (see Supplementary Note 9), which allow us to evaluate the $B$ dependence of $T_e - T_n$. As shown in Fig. 3b, it is maximal at zero field by the strong thermalization (i.e., $T_n \sim T_p$) and decreases gradually with $B$. There is a crossover field $B_c$, marked by $\Gamma_{np}$ falling below $\Gamma_{ne}$ (see the results at $T = 100$ mK and 1 K in Fig. 3b). In Fig. 3c, we compare the $B$ dependence of the experimental $V/I^2_{rms}$ (blue plots) for Device 2 and calculated $V/I^2_{rms}$ based on the nuclear SSE $\mathcal{S}_n$ (red solid curve) at $T = 100$ mK. Of important note, the experimental data are quantitatively reproduced by the calculation. Such agreement is confirmed also for other $B$ and $T$ regions (see Fig. 3d, e). A non-monotonic field response of $V$ now becomes evident: for $B \ll B_c$, the SSE signal increases in proportion to $B$ ($\mathcal{S}_n \propto B$), owing to the increased canting angle, and it takes a maximum at $B \sim B_c$. For $B \gg B_c$, the SSE signal decreases monotonically with $B$ ($\mathcal{S}_n \propto B^{-1}$) due to the reduction of thermal nonequilibrium $T_e - T_n$ ($\propto B^{-2}$) between the electron and nuclear systems (see Fig. 3b). We also evaluated the electronic SSE, $\mathcal{S}_m$, driven by the antiferromagnetic spin-wave mode $\omega_{mk}$ (see Supplementary Note 9 for details) and found that its intensity, as well as $B$ and $T$ dependencies, do not explain the experimental results (see Fig. 3c and its inset), which confirm that the nuclear SSE dominates the observed SSE.

**Discussion**

We finally discuss the difference between the previous nuclear-spin pumping[28] and the present nuclear SSE. For the nuclear-spin pumping, the measured voltage is maximal at a relatively low field of ~0.3 T and then starts to decrease with $B$. In such a low-$B$ range, the excitation gap of electronic spin-wave branch in MnCO₃ is comparable to that of the nuclear spins, and a nuclear-spin wave, hybridized electronic spin-wave and nuclear-spin mode[32–34], is excited. The experimental result in ref. 28 was thereby attributed to the coherent nuclear spin-wave formation, the electronic (magnetization) component of which pumps a spin current into an adjacent metallic layer in analogy with the

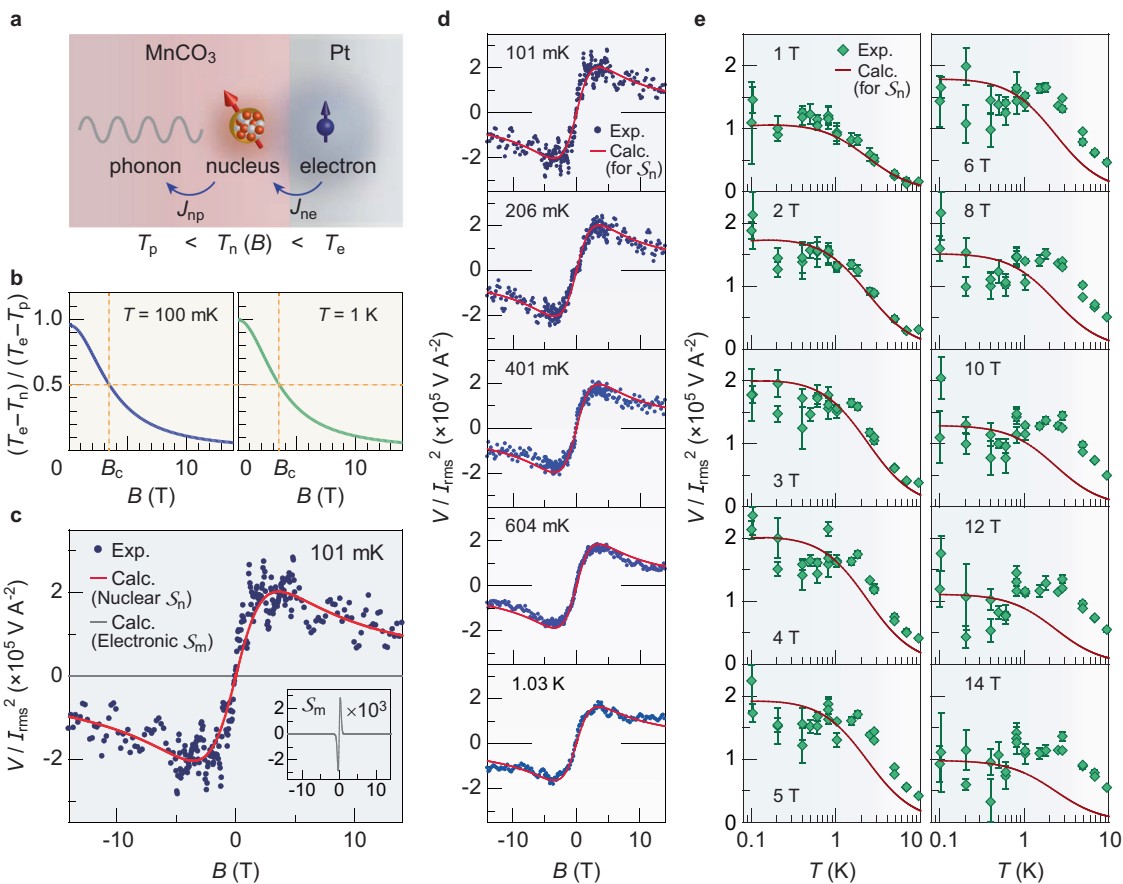

**Fig. 3 Comparison between experiment and theory. a** Interfacial nuclear-spin current and thermal equilibration of nuclear spins in $MnCO_3$. An interfacial spin current, $J_{ne}$, is mediated by the Korringa process through the hyperfine interaction between nuclear spins of $^{55}Mn$ and electron spins in the metal at the $Pt/MnCO_3$ interface. $J_{ne}$ arises in proportion to the effective temperature difference between the electrons in Pt ($T_e$) and nuclei in $MnCO_3$ ($T_n$): $J_{ne} = \Gamma_{ne} k_B (T_e - T_n)$. Here, the difference $T_e - T_n$ may be triggered by the interfacial temperature drop $T_e - T_p$ between the Pt and $MnCO_3$ ($T_p$: phonon temperature in $MnCO_3$ close to the interface) and the thermalization between nuclei and phonons in $MnCO_3$, whose rate is given by $J_{np} = \Gamma_{np} k_B (T_n - T_p)$. **b** $B$ dependence of the calculated temperature difference $T_e - T_n$ normalized by the interfacial temperature drop $T_e - T_p$ at $T = 100$ mK and 1 K. In the steady state, $J_{ne} = J_{np}$[31], which gives $(T_e - T_n)/(T_e - T_p) = \Gamma_{np}/(\Gamma_{np} + \Gamma_{ne})$. $B_c$ indicates the crossover field, where $\Gamma_{ne} = \Gamma_{np}$. **c** Comparison between the $B$ dependence of the experimental $V/I^2_{rms}$ (blue plots) for the $Pt/MnCO_3$ Device 2 and the calculated $V/I^2_{rms}$ for the nuclear SSE $S_n$ (red solid curve) and for the electronic SSE $S_m$ (gray solid curve) at $T = 101$ mK (see Supplementary Note 9 for details). The inset shows a blowup of the calculated $V/I^2_{rms}$ for the electronic SSE $S_m$ (multiplied by $10^3$). **d** Comparison between the $B$ dependence of the experimental $V/I^2_{rms}$ (blue plots) and the calculated $V/I^2_{rms}$ for the nuclear SSE $S_n$ (red solid line) at 100 mK < $T$ < 1 K. **e** Comparison between the $T$ dependence of the experimental $V/I^2_{rms}$ (green rhombus) and the calculated $V/I^2_{rms}$ for the nuclear SSE $S_n$ (red solid curve). The error bar represents the SD.

conventional electronic spin pumping for a magnet/metal bilayer. On the other hand, the present nuclear SSE increases with $B$ up to around 4–5 T, whereas nuclear-electronic hybridization is quickly suppressed as the electronic spin waves become gapped out. This suggests that a different physical mechanism governs the nuclear SSE, which is reasonable, as the nuclear pumping in the SSE is not limited to a coherent long-wavelength dynamics. We thus develop a nuclear SSE theory in terms of interfacial Korringa relaxation, in which nuclear-spin fluctuation directly transmits a spin current into an attached metallic layer via interfacial hyperfine interaction, and found quantitative agreement between the experiment and calculation. The Korringa mechanism does not need strong nuclear-electronic spin hybridization in the magnetic layer and also electronic spin transfer at the interface. This may extend a class of materials applicable for nuclear spintronics; materials having magnetic elements with nuclear spins and strong hyperfine interaction, such as $^{55}Mn$ and $^{59}Co$ (both of which are 100% natural abundance), can be potential sources of nuclear-spin currents.

In summary, we demonstrated the thermoelectric conversion driven by nuclear spin: the nuclear SSE. The nuclear SSE is enhanced at ultralow temperatures, in stark contrast to conventional electron-based thermoelectricity. It is surely worthwhile to explore nuclear SSEs in other systems to show the generality of the phenomenon. Materials of interest include easy-axis anti-ferromagnetic insulators having a large nuclear spin and exhibiting a spin-flop transition, at which the electronic magnon gap comes close to the low energy scales relevant to the nuclear dynamics[35]; for the nuclear SSE, this is instrumental in thermal equilibration of the nuclei within the magnetic material.

The present work may serve as the bridge between nuclear-spin science and thermoelectricity and marks the beginning of a research field "Nuclear thermoelectricity". It is also worth exploring the reciprocal of the nuclear SSE, as it will be applied to making a nuclear heat pump working at ultralow temperatures.

## Methods
**Sample preparation**. We used single-crystalline $MnCO_3$ slabs with a size of $3 \times 3 \times 0.5$ mm$^3$, which are commercially available from SurfaceNet. The largest plane is

(111) in the rhombohedral representation[36,37]. On the top of the (111) plane of the $MnCO_3$ slabs, 10 nm-thick Pt strips (200 μm long and nominally 100 nm wide) were patterned by electron beam lithography and lift-off methods (see also Supplementary Note 2). The Pt strips were deposited by magnetron sputtering in a $10^{-1}$ Pa Ar atmosphere. For a control experiment, we also prepared $W/MnCO_3$ devices, where the Pt strips are replaced with 10 nm-thick W strips (200 μm long and 500 nm wide) exhibiting a negative $\theta_{SHE}$[10,38,39].

**SSE measurement**. We measured the SSE by a standard lock-in technique[11,14,40,41] with a PPMS (Quantum Design) from 1.8 to 50 K and a $^3$He–$^4$He dilution refrigerator (KelvinoxMX200, Oxford Instruments; cooling power of 200 μW at 100 mK) from 100 mK to 10 K. An a.c. charge current ($I_c = \sqrt{2}I_{rms} \sin \omega t$) was applied to the Pt strip with a current source (6221, Keithley) and the generated voltage $V$ across the strip was recorded with a lock-in amplifier (LI5640, NF Corporation). For the measurements with the dilution refrigerator, we further introduced a voltage preamplifier (1201, DL Instruments) and a programmable filter (3625, NF Corporation) to reduce signal noise. The typical a.c. charge current property is as follows: the root-mean-square (rms) amplitude $I_{rms}$ of 0.1–5 μA and the frequency $\omega/2\pi$ of 13.423 Hz. All the $V$–$B$ data are anti-symmetrized with respect to the magnetic field $B$.

## Data availability
The data that support the findings of this study are available from the corresponding author upon reasonable request.

## Code availability
The codes used in theoretical simulations and calculations are available from the corresponding authors upon reasonable request.

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

## Acknowledgements

We thank Y. Chen, J. Lustikova, T. Hioki, N. Yokoi, H. Chudo, M. Imai, K. Sato, and G. E. W. Bauer for fruitful discussions and T. Nojima for his valuable comments on low-temperature experiments. This work was supported by JST ERATO "Spin Quantum Rectification Project" (JPMJER1402), JST CREST (JPMJCR20C1 and JPMJCR20T2), JSPS KAKENHI (JP19H05600, JP19K21031, JP20H02599, JP20K22476, and JP20K15160), MEXT [Innovative Area "Nano Spin Conversion Science" (JP26103005)], and Daikin Industries, Ltd. The work at UCLA was supported by the US Department of Energy, Office of Basic Energy Sciences under Award number DE-SC0012190. K.O. acknowledges support from GP-Spin at Tohoku University. R.R. acknowledges support from the European Commission through the project 734187-SPICOLOST (H2020-MSCA-RISE-2016), the European Union's Horizon 2020 research and innovation program through the Marie Sklodowska-Curie Actions grant agreement SPEC number 894006 and the Spanish Ministry of Science (RYC 2019-026915-I).

## Author contributions

T.K., T.S., and K.O. fabricated the devices. T.K. and T.M. constructed the experimental setup with the help of R.R. and K.O. T.K., T.M., H.I., T.S., K.T., and S.D. performed the experiments and collected the data. T.K. and H.I. analyzed the data with input from D.R. D.R. and Y.T. developed the theoretical explanations. E.S. and Y.T. conceived and supervised the project. T.K. and D.R. wrote the paper with review and input from E.S. and Y.T. T.K., D.R., H.I., T.M., T.S., K.T., S.D., K.O., R.R., S.T., Y.S., Y.T., and E.S. discussed the results and commented on the manuscript.

## Competing interests
The authors declare no competing interests.
