## [Peer Review File · Nature Communications]

REVIEWERS' COMMENTS

Reviewer #2 (Remarks to the Author):

Report on manuscript

Authors: Kikkawa et al

Title: Observation of Nuclear-spin Seebeck effect

This manuscript reports experimental and theoretical studies of the spin Seebeck effect in MnCO_3 , an antiferromagnetic insulator that is known to have nuclear spin and sizeable hyperfine coupling. The experimental data show that the voltage generated by a thermal gradient, as in the spin Seebeck effect, exhibit a behavior quite different than that observed in other magnetic compounds, namely, it increases instead of decreasing when the temperature is decreased. The authors interpret this behavior as due to nuclear spin-driven spin Seebeck effect, mediated by the large hyperfine interaction. This interpretation is supported by a theory that qualitatively agrees with the experimental data. In my opinion this is a new quite interesting phenomenon, that will be of interest to the community working in the areas of antiferromagnetic spintronics and spin caloritronics. The experiments and interpretation are presented in detail, and the paper is well written. I also believe that the author's replies to the comments and questions of the previous reviewers are satisfactory. Thus, I recommend acceptance for publication in Nature Communications.

[Authors' response to the Reviewer #2]

[Comment]

This manuscript reports experimental and theoretical studies of the spin Seebeck effect in MnCO₃, an antiferromagnetic insulator that is known to have nuclear spin and sizeable hyperfine coupling. The experimental data show that the voltage generated by a thermal gradient, as in the spin Seebeck effect, exhibit a behavior quite different than that observed in other magnetic compounds, namely, it increases instead of decreasing when the temperature is decreased. The authors interpret this behavior as due to nuclear spin-driven spin Seebeck effect, mediated by the large hyperfine interaction. This interpretation is supported by a theory that qualitatively agrees with the experimental data. In my opinion this is a new quite interesting phenomenon, that will be of interest to the community working in the areas of antiferromagnetic spintronics and spin caloritronics. The experiments and interpretation are presented in detail, and the paper is well written. I also believe that the author's replies to the comments and questions of the previous reviewers are satisfactory. Thus, I recommend acceptance for publication in Nature Communications.

[Our reply]

We thank the Reviewer for the valuable time spent on the evaluation of our manuscript and the recommendation to publish it in Nature Communications.